# Reconstruction of OFDM Signals Using a Dual Discriminator CGAN with BiLSTM and Transformer

**DOI:** 10.3390/s24144562

**Published:** 2024-07-14

**Authors:** Yuhai Li, Youchen Fan, Shunhu Hou, Yufei Niu, You Fu, Hanzhe Li

**Affiliations:** 1Graduate School, Space Engineering University, Beijing 101416, China; yuhai_li@hgd.edu.cn (Y.L.); hshhsc2022@163.com (S.H.); fionanyf@163.com (Y.N.); you_fu@hgd.edu.cn (Y.F.); hance0601@outlook.com (H.L.); 2School of Space Information, Space Engineering University, Beijing 101416, China

**Keywords:** communication signal, signal reconstruction, orthogonal frequency division multiplexing, LSTM, transformer, conditional generative adversarial network

## Abstract

Communication signal reconstruction technology represents a critical area of research within communication countermeasures and signal processing. Considering traditional OFDM signal reconstruction methods’ intricacy and suboptimal reconstruction performance, a dual discriminator CGAN model incorporating LSTM and Transformer is proposed. When reconstructing OFDM signals using the traditional CNN network, it becomes challenging to extract intricate temporal information. Therefore, the BiLSTM network is incorporated into the first discriminator to capture timing details of the IQ (In-phase and Quadrature-phase) sequence and constellation map information of the AP (Amplitude and Phase) sequence. Subsequently, following the addition of fixed position coding, these data are fed into the core network constructed based on the Transformer Encoder for further learning. Simultaneously, to capture the correlation between the two IQ signals, the VIT (Vision in Transformer) concept is incorporated into the second discriminator. The IQ sequence is treated as a single-channel two-dimensional image and segmented into pixel blocks containing IQ sequence through Conv2d. Fixed position coding is added and sent to the Transformer core network for learning. The generator network transforms input noise data into a dimensional space aligned with the IQ signal and embedding vector dimensions. It appends identical position encoding information to the IQ sequence before sending it to the Transformer network. The experimental results demonstrate that, under commonly utilized OFDM modulation formats such as BPSK, QPSK, and 16QAM, the time series waveform, constellation diagram, and spectral diagram exhibit high-quality reconstruction. Our algorithm achieves improved signal quality while managing complexity compared to other reconstruction methods.

## 1. Introduction

Communication signal reconstruction technology is a crucial research focus in communication countermeasures [1] and signal processing [2]. In recent years, achieving high-fidelity signal reconstruction has become increasingly challenging with the increasing complexity of the electromagnetic environment. Consequently, there is an urgent need for research into signal reconstruction technology based on new advancements and innovative concepts.

When confronted with unknown communication signals, traditional methods are predominantly reconstructed through the subsequent procedures. Initially, blind estimation is used to determine the signal parameters, including modulation style, symbol rate, and filter parameters. Subsequently, the channel parameters are estimated. Finally, based on these estimations, the signal and channel parameters are acquired, followed by the reconstruction of the signal using conventional modulation techniques. However, when faced with complex communication protocols and digital signal styles, the traditional blind parameter estimation methods struggle to estimate and reconstruct accurately.

In particular, for OFDM modulation system signals, the existing traditional blind estimation algorithms primarily focus on blind estimation of parameters such as the modulation mode [3], carrier frequency [4], chip timing width [5], and cyclic prefix length [6]. Blind parameter estimation of OFDM signals in a complex electromagnetic environment still presents challenges, including susceptibility to noise and imprecise estimation [7], intricate estimation procedures, high computational complexity [8], and significant parameter estimation errors. As a result, traditional blind estimation schemes are not directly applicable to OFDM signal reconstruction.

In the past few years, deep learning technologies have been widely applied in signal processing [9,10]. The emergence of generative adversarial networks [11] and other generative technologies has introduced new methods and concepts for complex communication signal reconstruction technologies.

In the field of single-carrier digital signal reconstruction, Yang et al. [12] effectively applied BEGAN (Boundary Equilibrium Generative Adversarial Networks) to reconstruct single-carrier BPSK and 8PSK signals. They evaluated the reconstructed signal quality by comparing it with the original real signal in various aspects such as symbol rate, spectrum characteristics, and constellation diagram. However, it should be noted that this method faces challenges in achieving signal reconstruction at higher sampling points.

Feng et al. [13] introduced DRAGAN, a CNN-based reconstruction network designed for high-precision reconstruction of complex signals with specific frame structures. Experimental results showed that DRAGAN captures modulation style, symbol rate, frequency bandwidth, and other key features effectively. However, its evaluation of reconstructed signal quality relies solely on the mean value and standard deviation, lacking a comprehensive similarity analysis system.

In general, while GAN-based communication signal reconstruction methods demonstrate promising results in reconstructing simple single-carrier communication signals, they face challenges and difficulties when it comes to reconstructing OFDM communication signals due to their noise-like characteristics in the time domain [14].

There is limited research on the reconstruction of OFDM signals. Due to the unique characteristics of OFDM signals, deep learning algorithms are unable to capture the noise-like waveform of OFDM directly. LIN et al. [14] proposed a method to convert the time-frequency two-dimensional pattern of OFDM signals into a GAN network for reconstruction and to learn the pixel information in a time-frequency graph. However, this approach requires excessive computation and involves converting the signal into an image before learning, making it more complex and requiring prior signal parameter information of OFDM signal.

In our previous research, we presented a method for directly reconstructing OFDM sequential sampled signals [15], established an OFDM signal reconstruction model, and proposed an OFDM signal sequential data reconstruction algorithm based on Transformer. Our approach demonstrated good refactoring quality under the two modulation styles of BPSK and QPSK. However, we observed that when confronted with more complex modulation methods, such as 16QAM, the reconstruction effect was unsatisfactory, and it became challenging to recover the constellation map information.

Therefore, the main contributions of this paper are as follows:We propose a novel algorithm for reconstructing OFDM signal timing sequence data, integrating the BiLSTM (Bi-directional Long Short-Term Memory) and Transformer models within a dual-discriminator CGAN architecture. This approach effectively extracts complex temporal information and accurately captures correlations among IQ signal components, achieving high-precision OFDM signal reconstruction;Utilizing an enhanced CGAN framework, we trained the network on data from all modulation format categories to learn comprehensive OFDM signal characteristics. This enables the generation of OFDM signals with different modulation categories by adjusting input parameters;Experimental results demonstrate that under common OFDM modulation schemes like BPSK (Binary Phase Shift Keying), QPSK (Quadrature Phase Shift Keying), and 16QAM (16-Quadrature Amplitude Modulation), the reconstructed signals closely resemble the original OFDM signals in time-domain waveforms, constellation diagrams, and spectrograms. Furthermore, the algorithm maintains complexity while improving the quality of the reconstructed signal compared to other reconstruction algorithms.

## 2. Background

### 2.1. Conditional Generative Adversarial Network

GAN is an unsupervised learning model proposed by Goodfellow et al. [11] in 2014, consisting of two structurally-independent deep learning networks: the generator and the discriminator. By training alternately between the generator and the discriminator, they eventually reach a Nash equilibrium. The data generated by the generator approximate the distribution of real data as closely as possible, making it difficult for the discriminator to distinguish between real samples and generated data.

The CGAN [16] is a specialized form of the GAN, incorporating improvements to combine supervised and unsupervised learning techniques, introducing conditional variables to guide the generation process, and adding conditional values to both the generator and discriminator inputs. This allows the CGAN generator to learn the mapping relationship of sample probability distribution under corresponding conditions.

Therefore, the CGAN was chosen as the foundational model for the network, allowing for simultaneous input of signals with diverse modulation types during training. This facilitates the effective capture of modulation information in the shallow layer of the signal. By manipulating label variables, we can induce the generator to reconstruct various signal types. The structural depiction of CGAN is presented in Figure 1.

The objective optimization function of CGAN is formulated as follows:(1)$$\underset{G}{\min}\underset{D}{\max}V\left(D,G\right)=E_{x∼p_{data}\left(x|c\right)}\left[\log\left(D\left(x,c\right)\right)\right]+E_{z∼p_{z}\left(z\right)}\left[\log\left(1-D\left(G\left(z,c\right),c\right)\right)\right]$$
(2)$$\underset{D}{\max}V\left(D,G\right)=E_{x∼p_{data}\left(x|c\right)}\left[\log\left(D\left(x,c\right)\right)\right]+E_{z∼p_{z}\left(z\right)}\left[\log\left(1-D\left(G\left(z,c\right),c\right)\right)\right]$$

The input *z* follows the random distribution $p_{z}\left(z\right)$, V(G,D) is the combined loss function of GAN, G(z,c) represents the data generated by *G*, D(x,c) and D(G(z,c),c) represent the probability that *D* gives the correct judgment for the real data and the generated data, respectively, where *c* is the added conditional information.

### 2.2. Transformer

The Transformer model architecture, proposed by Ashish Vaswani et al. [17] in 2017, is characterized by its self-attention mechanism, allowing for efficient capture of global data relationships during the transformation of input features into output features. This feature has contributed to the remarkable success of Transformer in the field of NLP (Natural Language Processing).

In light of the robust presentation capabilities of Transformer, numerous network models based on its architecture have been proposed by researchers in recent years. The most notable models in the field of computer vision include VIT (Vision in Transformer) [18], Swin Transformer [19], and other similar models. These models have demonstrated outstanding performance in tasks such as image classification [20], object detection [21], image generation [22], and others.

The core concept of VIT is to transform images into a format suitable for the Transformer architecture originally designed for NLP tasks. It achieves this by dividing the image into smaller patches, analogous to words in a sentence. Each patch undergoes Patch Embedding, where a fully connected network compresses it into a fixed-dimensional vector. Positional information is then incorporated into these embeddings. VIT adapts the Transformer’s encoder architecture, which includes layer normalization, multi-head attention, drop path, another layer normalization, a multi-layer perceptron, and another drop path, providing a tailored approach for image processing tasks.

Based on image processing models, Transformer-based GAN networks have also thrived. In 2022, Google introduced a network architecture called VITGAN [23], the first model to utilize Vision Transformer for GAN training. Inspired by VIT and BERT structures, VITGAN divides the input image into pixel blocks of size patch_size × patch_size and feeds them into the Transformer-based discriminator for training. Simultaneously, the generator transforms the noise z into a latent vector w and injects it into the network for training. After generating pixel blocks, the generator combines them to form a complete image. VITGAN has demonstrated comparable performance to CNN models, opening up new possibilities in image generation.

In the same year, Zhang et al. [24] proposed a StyleSwin-GAN model based on the foundational framework of the Swin Transformer for high-resolution image generation. To exploit the local and shifted window contexts, StyleSwin works based on a double attention mechanism and showcases superior performance compared to all existing Transformer-based GANs, demonstrating the potential of Transformers in generating high-resolution images.

In addition to its extensive application in computer vision, Transformer has demonstrated remarkable effectiveness in time series prediction, classification, generation, and other tasks due to its exceptional capability to capture long-term dependencies within the realm of time series data processing.

In the field of time series forecasting, Cao et al. [25] introduced a novel hybrid architecture the LSTM-Transformer designed specifically for multi-task real-time forecasting. This model integrates the key strengths of both LSTM and Transformer architectures to dynamically adapt to varying operating conditions and continuously assimilate new field data through online learning and knowledge distillation technology.

In the field of time series classification, Sera Kim et al. [26] proposed a new emotion recognition model architecture that combines the BiLSTM Transformer and two-dimensional convolutional neural networks. The BiLSTM-Transformer is used to process audio features to capture sequences of speech patterns, while the 2D CNN is used to process Mayer spectrographs to capture spatial details of audio.

In the field of time series data generation, Li [27,28] innovatively applied the Transformer model. Inspired by VIT, multiple one-dimensional time series signals are treated as multi-channel image data and fed into a network similar to VITGAN [23] for training, yielding promising results.

With the ongoing advancement of Transformer technology in image and timing signal generation, this study aims to integrate these GAN-based Transformer concepts into the realm of communication signal reconstruction.

## 3. Establishment of an OFDM Signal Dataset

In the investigation of OFDM communication signals, due to the limited availability of publicly accessible dataset resources, this study opted to employ simulation for generating datasets in order to facilitate subsequent reconstruction of OFDM signals.

### 3.1. OFDM System Simulation

The OFDM communication system [29] was constructed using MATLAB, and the system block diagram is illustrated in Figure 2. Initially, a random sequence of 0 and 1 was generated from the source, followed by serial-to-parallel conversion post-modulation. Subsequently, IFFT was performed after inserting the pilot frequency to obtain the time-domain signal, followed by adding cyclic prefix and parallel-to-serial conversion before passing through the multipath channel. At the receiving end, serial-to-parallel, cyclic prefix removal, FFT, channel equalization, demodulation, and parallel-to-serial were carried out to obtain the received information sequence.

### 3.2. Dataset Details

To simulate signal interception in a realistic communication countermeasure environment, we selected the received signal *y*(*t*) after transmission through the channel to construct our OFDM dataset. The modulation formats of BPSK, QPSK, and 16QAM commonly used in OFDM communication were included. However, achieving an ideal high SNR during signal reconstruction may deviate from reality, while a low SNR can drown useful features in noise and hinder learning. Therefore, we selected three SNR levels (10 dB, 15 dB, and 20 dB) with 2000 signals for each modulation style under each SNR level, resulting in a total of 18,000 signal data points. The details of the dataset are presented in Table 1.

## 4. LSTM and Transformer-Based Dual-Discriminator CGAN Model Architecture

### 4.1. OFDM Signal Reconstruction Model

The OFDM signal reconstruction model utilized in this study was built on our previous work [15]. As the OFDM signal in the time domain comprises multiple orthogonal subcarriers, it adheres to a Gaussian distribution with zero mean [14] according to the central limit theorem. Possessing characteristics akin to noise proves challenging for neural networks to extract features from it. Conversely, OFDM frequency domain signals exhibit more prominent features and are better suited for neural network learning. Consequently, based on the modulation process of OFDM, a reconstruction process was devised to acquire its frequency domain information.

Initially, we conducted channel equalization operations on the *y*(*t*) signals within the constructed dataset to mitigate the impact of frequency offset in the signals. Subsequently, Serial/Parallel conversion, Cyclic Prefix removal, and FFT pre-processing operations were carried out based on known partial prior information. This resulted in obtaining a frequency domain signal with distinct characteristic information, which was then fed into the GAN network for learning and training. The reconstructed signal exhibited high similarity to the original frequency domain signal. Finally, IFFT transformation, Cyclic Prefix insertion, and Parallel/Serial conversion were applied to recover the corresponding OFDM signals. The schematic diagram of the signal reconstruction model is illustrated in Figure 3.

### 4.2. Conditional GAN Model

One limitation of TOR-GAN [12], as proposed in our previous work, is its incapacity to concurrently train multiple modulation types of data. Consequently, a dataset comprising N classes necessitates the separate training of N distinct models and restricts independent reconstruction solely to each class of signal.

After observing that the Transformer GAN was unable to generate coherent reconstructed signals using traditional tag embedding methods in the generator and discriminator, we were motivated by previous research [28,30] to adopt a novel approach for integrating label information into GAN models. This involved the removal of labels from the real data used by the discriminator, exclusive embedding of labels in the noise generated by the generator, and integration of a classification header on the discriminator. Consequently, this enabled the generator to generate diverse classes of signal data while training the discriminator not only to differentiate between real and reconstructed signals but also to classify them according to their respective classes, as illustrated in Figure 4 of our model architecture.

### 4.3. Signal Reconstruction Network Structure

The extraction of time dimension information and the accurate capture of correlation between two IQ signals are crucial for reconstructing communication signals. Nevertheless, traditional time series processing networks are susceptible to gradient disappearance or explosion when handling long series OFDM signal data, posing challenges in accurately capturing the correlation between IQ signals.

LSTM effectively captures long-term dependencies in sequential data through a gating mechanism to mitigate the issues of vanishing or exploding gradients, providing significant advantages in processing time series data with long-term dependencies. Meanwhile, Transformer comprehensively captures sequence dependencies utilizing self-attention and multi-head attention mechanisms, enabling simultaneous consideration of all positions in the input sequence for better contextual understanding and efficient parallel computing. Furthermore, the concept of VIT can also incorporate signal reconstruction to better capture the correlation between two IQ signals. Consequently, we developed our signal reconstruction GAN with Transformer serving as its core network.

#### 4.3.1. Discriminator Network

As depicted in Figure 5, we constructed a dual discriminator network model with Transformer serving as the fundamental architecture. The idea for the construction of the discriminator was derived from [15,26]. The primary function of the first discriminator was to extract temporal characteristic information from IQ signals, and its specific parameters are detailed in Table 2. Initially, for direct input IQ time series, the AP feature of the signal was amalgamated into its temporal feature, resulting in an output size of 4 × 256 to ensure that the model acquired not only the timing feature but also the corresponding modulation constellation feature. These combined features were subsequently input into the BiLSTM network model to capture the long-term dependencies and relationships across different dimensions, resulting in an output size of 256 × Embed_size, which is twice the size of Hidden_dim in BiLSTM. Finally, the fixed-position coding in the original Transformer was applied to the time series data following the initial feature extraction.

Subsequently, we present the network architecture of the second discriminator, which was primarily tasked with learning the correlation between the two IQ signals. The specific parameters for this discriminator are detailed in Table 3. The IQ signal was treated as a single channel comprising two sampling sequences, each consisting of 256 sampling points. We structured the original IQ data as image data and divided the two input sequences into 128 small image blocks using a 2 × 2 Conv2d, expanding their dimensions to Embed_size. Finally, we incorporated fixed-position coding, resulting in a final output of 128 × Embed_size.

The two encoded signals were then input into the core network, which was based on the Transformer encoder layer, for further processing. Specific parameters of the core network are listed in Table 4. Batch normalization was initially conducted in the core network to accelerate the training process and enhance model stability. Subsequently, the multi-head self-attention mechanism was employed to capture position relationships between each encoded data and extract dependency and structural information from the data. The standardized features were then passed into the MLP network for additional processing. The MLP network consisted of two fully connected layers with an Expansion = 4, and the GELU activation function was utilized among the layers to enhance the non-linear expression capability of the model. Subsequently, both the output of the self-attention mechanism and the MLP network underwent Dropout operations individually to prevent overfitting. The input and output were then combined through Residual Connection to obtain the final output result. The final output size varied in the first dimension according to different discriminators.

Finally, the discriminant module was introduced, where the input data were dimensionally reduced to facilitate subsequent processing. Following layer normalization, they were then passed to the fully connected layer for signal authenticity identification. The specific parameters of the identification module are listed in Table 5.

#### 4.3.2. Generator Network

As shown in Figure 6, Transformer served as the central component of our generator network, inspired by TTS-GAN [27]. The parameters are listed in Table 6. A Gaussian noise vector z was used as input, which was then transformed through a fully connected layer to match the dimension of the IQ signal x and the embedding vector. To ensure consistent location information in both IQ signals and maintain their relevance in subsequent learning processes, one sequence was separately encoded, and its information was then mapped to the other sequence, resulting in a final output size of 2 × 256 × Embed_size.

Subsequently, the encoded vector was input into the core Transformer encoder layer, which possessed a network structure identical to that of the discriminator. The specific parameters for this layer are detailed in Table 7. Following assimilation from the core network, Conv2d was employed to reduce the dimensions of the embedded vector in order to conform with the final output requirements, resulting in an output size of 2 × 256.

## 5. Experiment and Analysis

### 5.1. Training Details and Parameters

The OFDM signal dataset consisted of a total of 18,000 signal data, with each modulation style signal having 2000 signal data at each SNR level. Subsequently, the OFDM signal *y*(*t*) underwent processing by the signal reconstruction model detailed in Section 4.1. Following this processing, it was transmitted to the CGAN network. Subsequently, to simulate real communication scenarios, data with the same SNR were fed into the CGAN network to enable the network to learn both modulation feature information and timing features. The specific training parameters are detailed in Table 8.

During the training process, the generator received a Gaussian-distributed random vector of shape (Batch size × 200) as input. Simultaneously, the initial learning rate for both the generator and discriminator was set to 0.001. To enhance training effectiveness, we implemented a linear decay learning rate function, resulting in a gradual decrease in the learning rate as training progressed until it reached 0 at the end of training. The model was iteratively optimized during training based on the loss function value to enhance its performance.

Furthermore, to ensure the repeatability and reliability of the experimental results, Table 9 provides a detailed overview of the hardware and software environment utilized during the model training process.

### 5.2. Experimental Results

The performance of the trained CGAN network is evaluated in this section by comparing the time-domain waveform diagram, constellation diagram, OFDM sequence waveform, and OFDM spectrograms of the reconstructed signal, as well as employing a signal similarity evaluation method.

#### 5.2.1. Comparison and Analysis of Signal Reconstruction Results

Figure 7, Figure 8 and Figure 9 present a detailed analysis of the time-domain waveform and constellation diagrams depicting the reconstructed model signal under varying SNR conditions. By manipulating the input label type of CGAN, the model demonstrated its capability to generate signals with diverse modulation styles.

Figure 7 illustrates that at an SNR of 20 dB, the time-domain waveform of the reconstructed signal closely resembled the original signal. The figure clearly depicts eight distinguishable OFDM symbols, with the model effectively capturing two pilot signals on each symbol. Furthermore, in the constellation diagram, it can be observed that the reconstructed signal accurately reflected the modulation characteristics of the original signal. The distribution of the sample points around the constellation diagram and the DC subcarriers in the center closely resembled the distribution of the original signal, and the model accurately learned the pilot signal at (1, 0). These results collectively indicate the strong performance of the model in signal reconstruction under high SNR conditions.

Figure 8 illustrates that at an SNR of 15 dB, the reconstructed signal effectively captured the key characteristics of the original signal, with clear identification of symbol and pilot information in the waveform diagram. However, for the 16QAM modulation, the relatively disordered distribution of constellation points resulted in a blurring of features in the reconstructed signal. Conversely, under BPSK and QPSK modulation styles with more regular constellation distributions, maintenance of constellation map features was superior, with minimal noise points evident in the reconstructed constellation map leading to improved reconstruction effects.

From Figure 9, it can be observed that at an SNR of 10 dB, although the features in the original signal became difficult to distinguish due to noise interference, the corresponding feature information could still be recovered by reconstructing the signal in the time-domain waveform diagram. In the BPSK and QPSK modulation schemes, the constellation mapping of the reconstructed signal closely resembled that of the original signal, in accordance with its original distribution characteristics. However, in the case of the 16QAM modulation, it was challenging to discern the distribution features in the original signal, leading to a fuzzy and indistinguishable reconstruction.

#### 5.2.2. Comparison and Analysis of the Original OFDM Sequences

After providing the OFDM frequency domain signals processed by the signal reconstruction model to the network, we initially conducted IFFT on these reconstructed signals to re-modulate them back to the time domain. Subsequently, we added a cyclic prefix before and after them in order to obtain the complete reconstructed OFDM sequence. To visually compare the difference between the reconstructed signal and the original signal, we plotted IQ waveform diagrams for both the reconstructed OFDM sequence and the corresponding graph of the original OFDM sequence. Additionally, we assessed the quality of the reconstructed signal by creating spectrograms for both the reconstructed and original signals to ensure accuracy and effectiveness in the reconstruction process.

The results of the comparison between the reconstructed OFDM sequence and the original OFDM sequence are illustrated in Figure 10, Figure 11, Figure 12, Figure 13, Figure 14 and Figure 15. In the comparison of the OFDM time-domain waveform diagrams, the blue curve represents the actual waveform, while the red curve depicts the reconstructed waveform. It is evident that the OFDM time-domain waveform exhibited noise-like characteristics. However, both the real and imaginary waveforms of the reconstructed signal demonstrated a high degree of similarity with the original signal, indicating that the reconstructed sequence effectively captured the distribution of the original signal.

In the comparison of the spectrogram, the red curve represents the true waveform, while the blue curve depicts the reconstructed waveform. Under the SNR of 20 dB, it is evident that the reconstructed spectral graph closely approximated the structure of the original spectrum. The characteristics of subcarrier numbers were clearly discernible, and there was effective recovery of pilot signals at both the 7th and 14th subcarriers. However, as the SNR decreased, discerning effective feature information in the reconstructed spectrogram became challenging due to noise-induced disturbances in the original spectrogram. Nevertheless, the reconstructed spectrogram maintained consistent overall distribution characteristics with the original signal.

#### 5.2.3. Model Comparison and Similarity Evaluation

In terms of model parameter quantity, we selected three existing models for comparative analysis with our proposed model. According to the data in Table 10, it is evident that our model remained commensurate with the other models in relation to generator parameter count. With regard to time complexity, although our model exhibited a marginally higher level than the Pattern-Constellation dual GAN, and demonstrated a lower time complexity compared to the remaining two models. In terms of the number of discriminator parameters, our parameter count was comparable to that of TOR-GAN. However, we made a trade-off in time complexity. Compared to other models, there was a significant reduction in the number of discriminator parameters and a corresponding decrease in time complexity.

When assessing the similarity between the reconstructed and original signals, we utilized three indicators to ensure a comprehensive evaluation, as shown in Table 11. We employed two commonly used indexes for evaluating time series data similarity: MSE (Mean Squared Error) and MAE (Mean Absolute Error). Additionally, in order to more accurately measure signal similarity, we also incorporated the use of EVM (Error Vector Magnitude) to achieve a comprehensive assessment of the model’s reconstruction effectiveness. It is evident that, in addition to SNR = 15 dB, certain performance metrics of the TOR-GAN model exhibited slight improvements to our model in BPSK and QPSK modulation modes. Conversely, our model demonstrated superior performance across all other modulation types, with significant enhancements observed in all key indicators. These findings serve to unequivocally validate the robustness and reliability of our model.

In addition, with regard to the evaluation of signal similarity, we also incorporated probability density functions to assess the quality of reconstructed signals. Figure 16, Figure 17 and Figure 18 depict the probability distributions of real and imaginary components of various signals. It is evident that at an SNR of 20 dB, the probability distributions of reconstructed signals closely aligned with those of the original signals. However, this alignment diminished as the SNR decreased. This reduction in correlation was especially noticeable in the context of the 16QAM modulation.

## 6. Conclusions

In this study, we proposed a dual discriminator CGAN model based on LSTM and Transformer architectures to address the challenges posed by the complexity and suboptimal performance of existing OFDM signal reconstruction methods. The proposed discriminator model not only facilitates the extraction of temporal information from the reconstructed OFDM signal model but also effectively captures the correlation between the in-phase and quadrature components of the signals. The reconstructed signals demonstrate high-quality reconstruction in the time-domain waveform diagram, constellation diagram, OFDM sequence waveform, and OFDM spectrogram under the commonly utilized BPSK, QPSK, and 16QAM modulation schemes in OFDM signals. In the actual model deployment, the pre-trained network weights can be utilized to train and update parameters when there are changes in signal parameters, presenting promising practical applications.

## 7. Future Work

We successfully achieved the reconstruction of a 16QAM complex modulated signal, not only restoring its high-quality time-domain waveform but also ensuring the fidelity of the reconstructed constellation characteristics. Furthermore, our algorithm showed improved performance in reconstructing BPSK and QPSK, two commonly used OFDM signal modulations, compared to existing methods.

While our current proposed algorithm demonstrated strong performance in OFDM signal datasets with time series sampling points of 256, the primary challenge encountered by the existing model when confronted with time-domain signals featuring longer sample points was the substantial escalation in the number of parameters for the generator. This proliferation of parameter count resulted in a marked increase in both time complexity and space complexity of the algorithm, thereby exerting an impact on its efficiency. Hence, our future research endeavors will focus on the development of a more efficient and suitable reconstruction model tailored for processing long sequence signals with enhanced precision and efficacy.

## Figures and Tables

**Figure 1 sensors-24-04562-f001:**
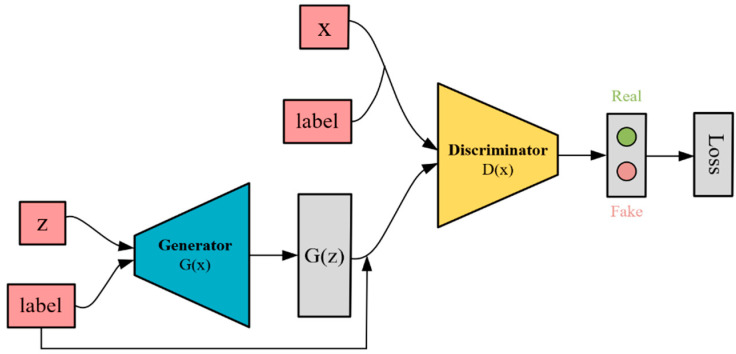
CGAN model architecture.

**Figure 2 sensors-24-04562-f002:**
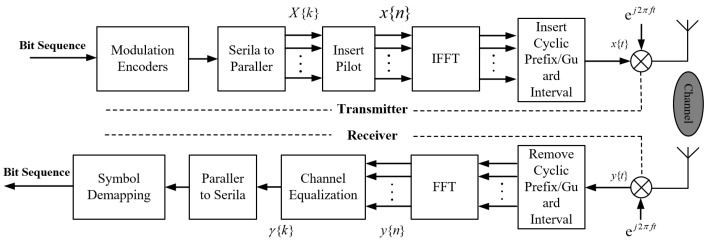
Flowchart of the OFDM baseband communication system.

**Figure 3 sensors-24-04562-f003:**
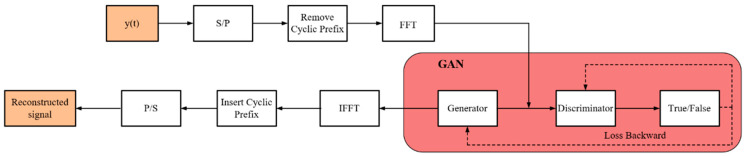
Signal reconstruction model.

**Figure 4 sensors-24-04562-f004:**
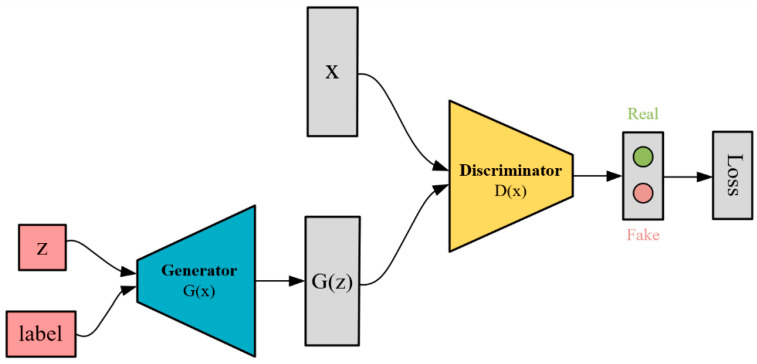
Improved CGAN model architecture.

**Figure 5 sensors-24-04562-f005:**
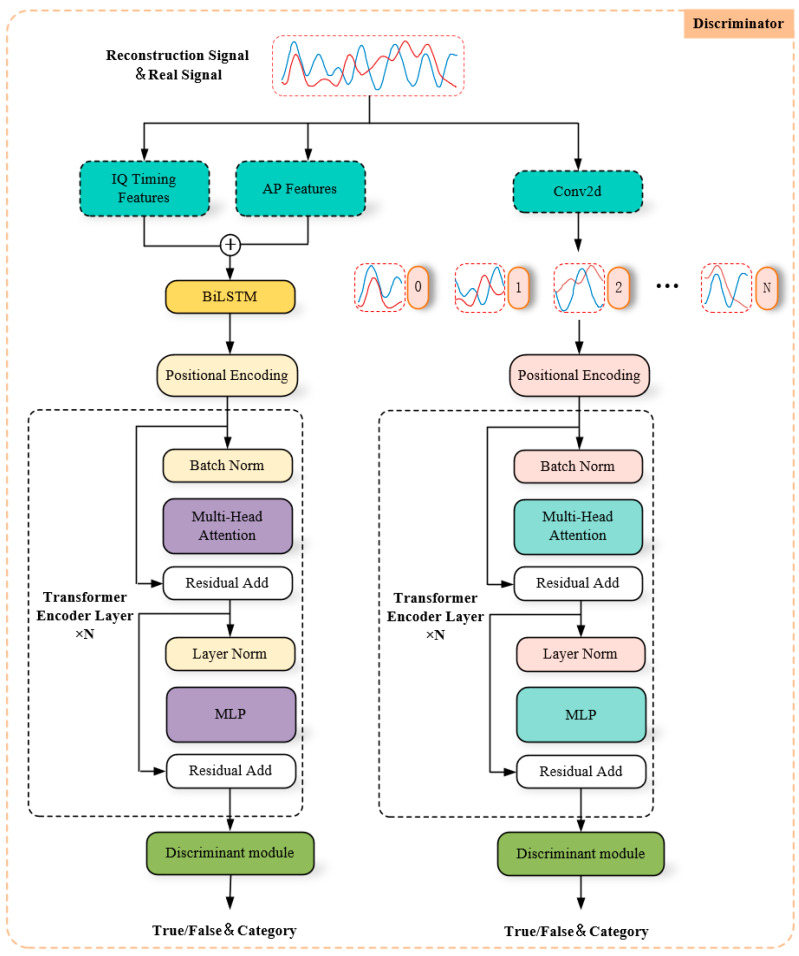
Discriminator network model.

**Figure 6 sensors-24-04562-f006:**
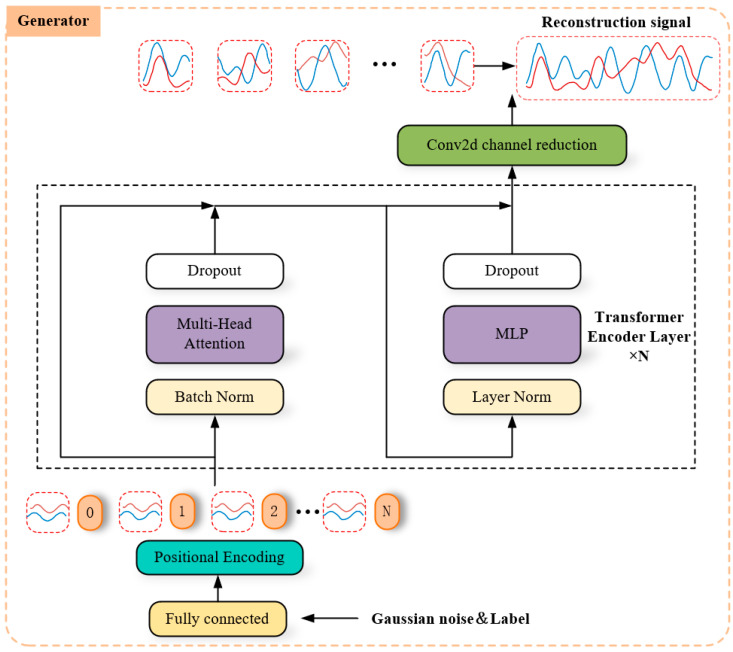
Generator network model.

**Figure 7 sensors-24-04562-f007:**
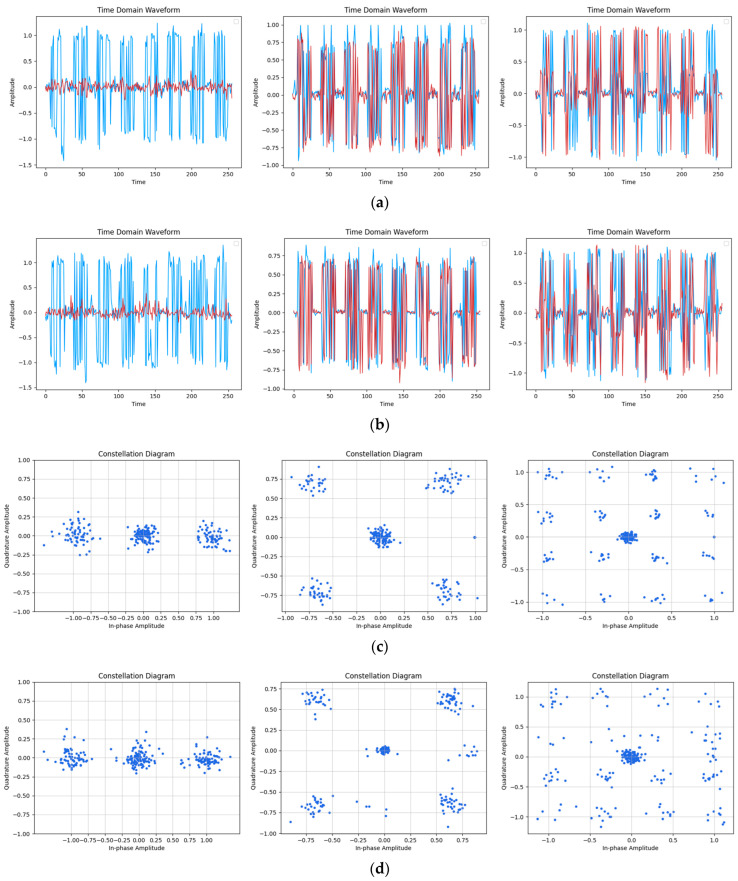
Reconstruction results of the time-domain waveform map and constellation diagram when SNR was 20 dB. (Left to right modulation style is BPSK, QPSK, 16QAM). (**a**) Original signal time-domain waveform; (**b**) Reconstruction of signal time-domain waveform; (**c**) Original signal constellation diagram; (**d**) Reconstruction of signal constellation diagram.

**Figure 8 sensors-24-04562-f008:**
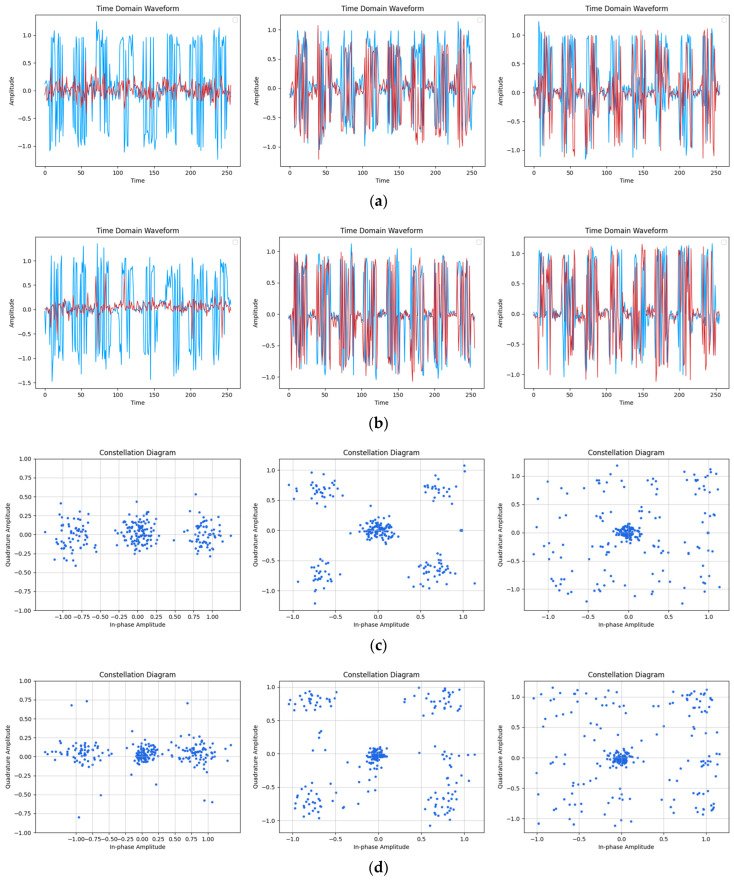
Reconstruction results of time-domain waveform map and constellation diagram when SNR was 15 dB. (Left to right modulation style is BPSK, QPSK, 16QAM). (**a**) Original signal time-domain waveform; (**b**) Reconstruction of signal time-domain waveform; (**c**) Original signal constellation diagram; (**d**) Reconstruction of signal constellation diagram.

**Figure 9 sensors-24-04562-f009:**
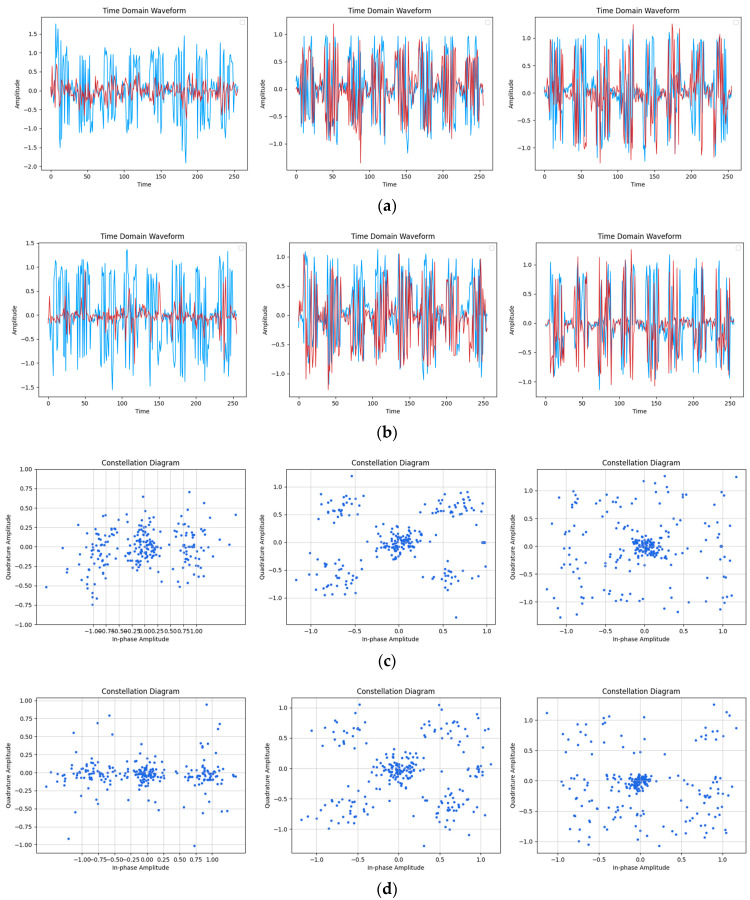
Reconstruction results of time-domain waveform map and constellation diagram when SNR was 10 dB. (Left to right modulation style is BPSK, QPSK, 16QAM). (**a**) Original signal time-domain waveform; (**b**) Reconstruction of signal time-domain waveform; (**c**) Original signal constellation diagram; (**d**) Reconstruction of signal constellation diagram.

**Figure 10 sensors-24-04562-f010:**
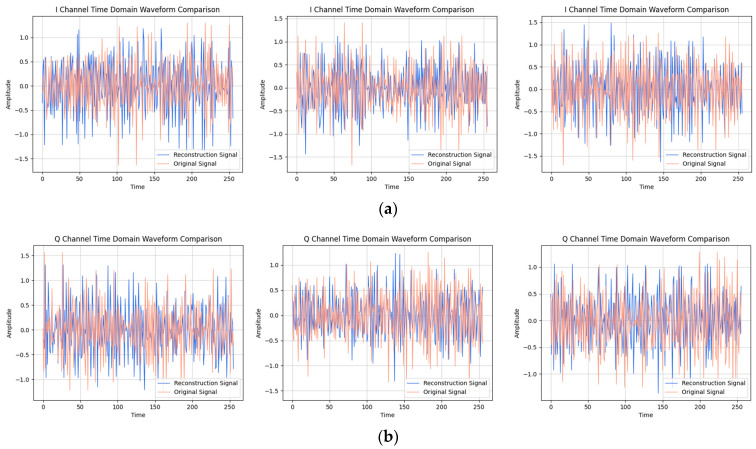
Visual comparison of time-domain waveform of OFDM symbol sequence when the SNR was 20 dB. (Left to right modulation style is BPSK, QPSK, 16QAM). (**a**) Real part; (**b**) Imaginary part.

**Figure 11 sensors-24-04562-f011:**
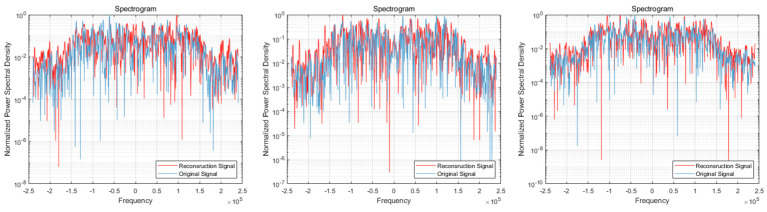
Visual comparison of OFDM signal spectrogram when the SNR was 20 dB. (Left to right modulation style is BPSK, QPSK, 16QAM).

**Figure 12 sensors-24-04562-f012:**
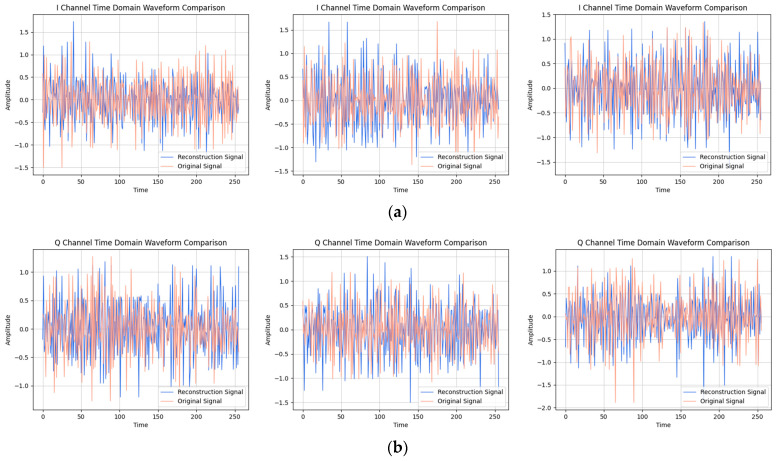
Visual comparison of time-domain waveform of OFDM symbol sequence when the SNR was 15 dB. (Left to right modulation style is BPSK, QPSK, 16QAM). (**a**) Real part; (**b**) Imaginary part.

**Figure 13 sensors-24-04562-f013:**
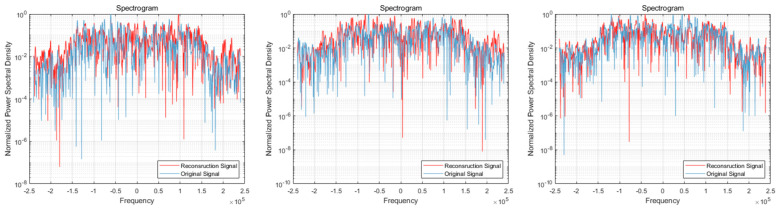
Visual comparison of OFDM signal spectrogram when the SNR was 15 dB. (Left to right modulation style is BPSK, QPSK, 16QAM).

**Figure 14 sensors-24-04562-f014:**
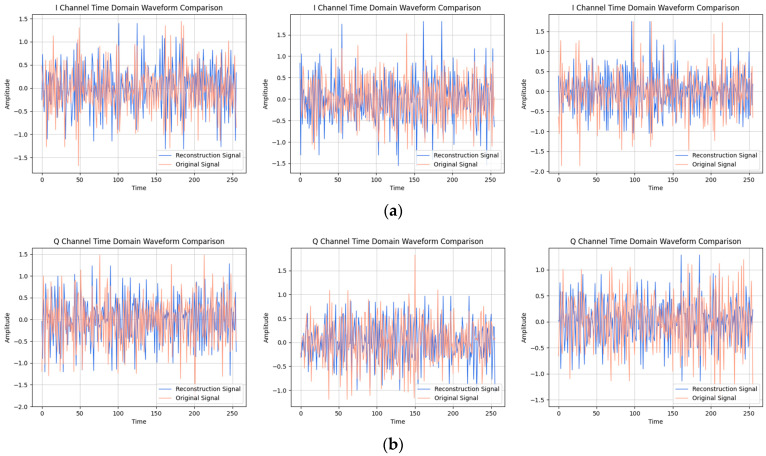
Visual comparison of time-domain waveform of OFDM symbol sequence when the SNR was 10 dB. (Left to right modulation style is BPSK, QPSK, 16QAM). (**a**) Real part; (**b**) Imaginary part.

**Figure 15 sensors-24-04562-f015:**
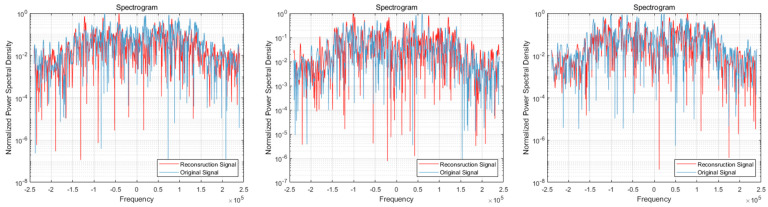
Visual comparison of OFDM signal spectrogram when the SNR was 10 dB. (Left to right modulation style is BPSK, QPSK, 16QAM).

**Figure 16 sensors-24-04562-f016:**
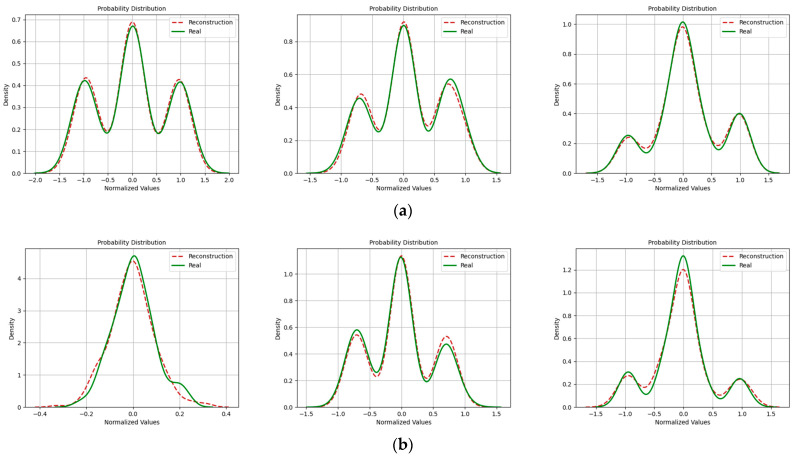
Probability density distribution when the SNR was 20 dB. (Left to right modulation style is BPSK, QPSK, 16QAM). (**a**) Real part; (**b**) Imaginary part.

**Figure 17 sensors-24-04562-f017:**
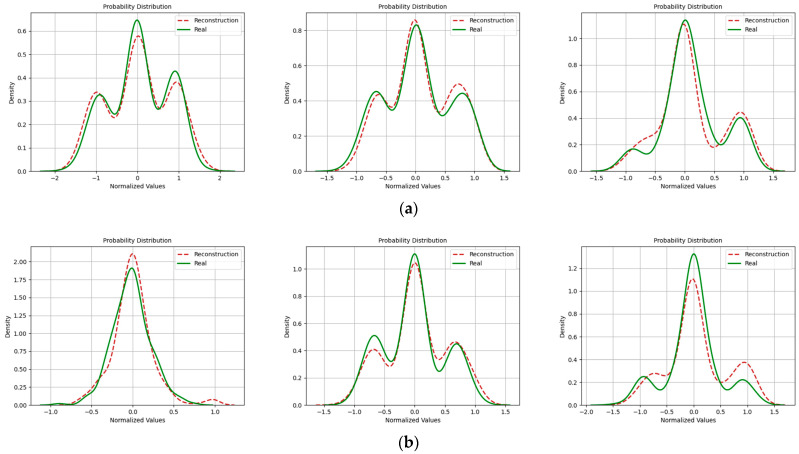
Probability density distribution when the SNR was 15 dB. (Left to right modulation style is BPSK, QPSK, 16QAM). (**a**) Real part; (**b**) Imaginary part.

**Figure 18 sensors-24-04562-f018:**
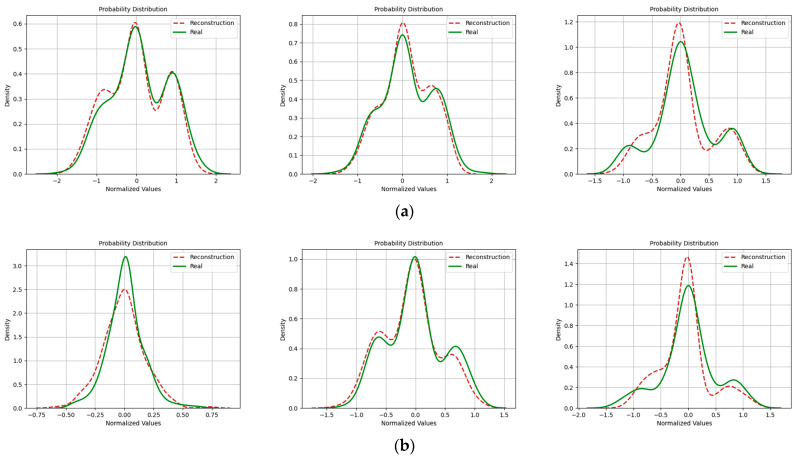
Probability density distribution when the SNR was 10 dB. (Left to right modulation style is BPSK, QPSK, 16QAM). (**a**) Real part; (**b**) Imaginary part.

**Table 1 sensors-24-04562-t001:** Dataset information.

Details	
Modulation mode	16QAM, QPSK, BPSK
Sample length	256
SNR range	(10 dB, 15 dB, 20 dB)
Signal sample size	18,000

**Table 2 sensors-24-04562-t002:** Network structure of the first discriminator.

Types	Paraments	Output Size
IQ + AP	-	4 × 256
BiLSTM	Num_layers = 1	256 × Embed_size
Positional_Encoding	-	256 × Embed_size

**Table 3 sensors-24-04562-t003:** Network structure of the second discriminator.

Types	Size/Step	Output Size
2D Convolution	2 × 2/2	128 × Embed_size
Positional_Encoding	-	128 × Embed_size

**Table 4 sensors-24-04562-t004:** Network structure of the discriminator Transformer encoder layer.

Types	Paraments	Output Size
BatchNorm 1D	Embed_size	256/128 × Embed_size
Multi-Head Attention	Num_heads = 16	256/128 × Embed_size
Dropout	Probability = 0.3	256/128 × Embed_size
Layer Normalization	Embed_size	256/128 × Embed_size
Linear	Expansion = 4	256/128 × 4 × Embed_size
GELU	-	256/128 × 4 × Embed_size
Linear	Expansion = 4	256/128 × Embed_size
Dropout	Probability = 0.3	256/128 × Embed_size

**Table 5 sensors-24-04562-t005:** Network structure of the discriminant module.

Types	Paraments	Output Size
Reduce	Reduction = mean	Embed_size
Layer Normalization	Embed_size	Embed_size
Linear	-	1

**Table 6 sensors-24-04562-t006:** Network structure of embedding.

Types	Paraments	Output Size
Linear	Laten_dim	2 × 256 × Embed_size
Positionl_Encoding	-	2 × 256 × Embed_size

**Table 7 sensors-24-04562-t007:** Network structure of the generator Transformer encoder layer.

Types	Paraments	Output Size
BatchNorm 1D	Embed_size	512 × Embed_size
Prob Attention	Num_heads = 16	512 × Embed_size
Layer Normalization	Embed_size	128 × Embed_size
Dropout	Probability = 0.3	512 × Embed_size
Linear	Expansion = 4	512 × 4 × Embed_size
GELU	-	512 × 4 × Embed_size
Linear	Expansion = 4	512 × 4 × Embed_size
Dropout	Probability = 0.3	512 × Embed_size
2D Convolution	Kernel_size = 1	2 × 256

**Table 8 sensors-24-04562-t008:** Network model training parameters.

Model Hyperparameters	
Generator Embed_dim	160
Discriminator Embed_dim	160
LSTM Hiddem_dim	320
Batch size	64
Epochs	75
Learning rate	0.001
*β*_1_, *β*_2_	0.9, 0.999
Optimizer	Adam

**Table 9 sensors-24-04562-t009:** Hardware and software environments.

Environment	Technical Parameters
OS	Windows 10
CPU	Intel Xeon Silver 4212R
GPU	NVIDIA GeForce 4090
Memory	128 G
Python	Python 3.8.8
Pytorch	Pytorch 1.8.1

**Table 10 sensors-24-04562-t010:** Comparison of model parameters.

Model Name	Generator		Discriminator	
	Parameters /B	Time Complexity /FLOPs	Parameters /B	Time Complexity /FLOPs
LSTM&Transformer Based CGAN	1.65 × 10^7^	3.42 × 10^8^	4.85 × 10^6^	1.97 × 10^8^
TOR-GAN [15]	1.69 × 10^7^	3.45 × 10^8^	5.37 × 10^6^	1.82 × 10^8^
Pattern-Constellation dual GAN [14]	1.65 × 10^7^	1.78 × 10^8^	2.04 × 10^7^	2.34 × 10^8^
DRAGAN [13]	1.43 × 10^6^	8.53 × 10^7^	6.99 × 10^7^	1.93 × 10^9^

**Table 11 sensors-24-04562-t011:** Reconstruction similarity evaluation.

Model Name	Similarity Analysis	BPSK			QPSK			16QAM		
		10 dB	15 dB	20 dB	10 dB	15 dB	20 dB	10 dB	15 dB	20 dB
LSTM&Transformer Based CGAN	MSE	0.3715	0.3129	0.1303	0.3008	0.2960	0.1520	0.3903	0.3224	0.2140
	MAE	0.3606	0.2731	0.1096	0.2975	0.2608	0.1403	0.4120	0.3055	0.1948
	EVM	1.1057	0.9635	0.7202	1.0359	0.9986	0.8134	1.3538	1.1489	1.0673
TOR-GAN	MSE	0.3970	0.3083	0.1460	0.4307	0.3086	0.1981	0.4624	0.3531	0.2529
	MAE	0.3413	0.2849	0.1284	0.4142	0.2981	0.1660	0.4546	0.2164	0.2502
	EVM	1.1395	0.9588	0.7911	1.2415	0.9935	0.8303	1.4145	0.9303	0.9137
DRAGAN	MSE	0.7051	0.5319	0.4197	0.6424	0.5740	0.4200	0.7135	0.6501	0.5265
	MAE	0.6376	0.4945	0.4086	0.5359	0.5542	0.3447	0.6995	0.5914	0.5209
	EVM	2.7872	1.5969	1.2761	2.4858	1.3715	1.1097	2.5895	1.4049	1.2388

## Data Availability

Data are contained within the article.

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
