# Peer review of "Reconstruction of OFDM Signals Using a Dual Discriminator CGAN with BiLSTM and Transformer"

_sensors, 2024, doi:10.3390/s24144562_

Round 1

Reviewer 1 Report

Comments and Suggestions for Authors

In this manuscript, the authors have demonstrated a novel algorithm for reconstructing OFDM signal timing sequence data, which had two deep learning models, LSTM and Transformer, and utilizes a dual-discriminator CGAN (Conditional Generative Adversarial Network) architecture. The algorithm could effectively extract intricate temporal information and accurately captures the correlation between the IQ signals, resulting in high-precision reconstruction of the OFDM signal. In my opinion, the text is well arranged and the logic is clear. However, there are several issues which need some revision before publication. The following questions should be responded reasonably.

1. Revise the “Abstract” and “ Introduction” since they are too lengthy.

2. The statement “Therefore, the main contributions of this paper are as follows: …” should be effectively downsized.

3. Some figures (for example, Figure 7 and Figure 8) have low resolution. Export the original and clear graph if possible.

4. Some references lack page numbers and volume numbers.

5. English expression and grammars need to be checked.

Comments on the Quality of English Language

English expression and grammars need to be checked.

Author Response

We have modified the article according to your comments, and the specific modification content is shown in word.

Reviewer 2 Report

Comments and Suggestions for Authors

The author only combines several mature existing technologies for signal reconstruction, and there is no substantial innovation in its content. At the same time, there are also the following problems:

What role does the BiLSTM network play in capturing timing details of the IQ sequence and constellation map information?

please discuss the experimental results in terms of the reconstruction quality of time series waveforms, constellation diagrams, and spectral diagrams?

How is the IQ sequence treated as a single-channel two-dimensional image, and how are pixel blocks segmented using Conv2d?

Comments on the Quality of English Language

Moderate editing of English language required

Author Response

(The authors gave the same response as above.)

Reviewer 3 Report

Comments and Suggestions for Authors

A LSTM and Transformer Based Orthogonal Frequency Division Multiplexing Signals Reconstruction Conditional GAN

by:

Yuhai Li, Youchen Fan, Shunhu Hou, Yufei Niu, You Fu, and Hanzhe Li.

This work is seems the continuity of their published work "TOR-GAN: A Transformer-Based OFDM Signals Reconstruction GAN".

The paper is nicely written.  They have proposed a dual discriminator CGAN model incorporating LSTM and Transformer.

I have the following observations:

1. Abstract contains the use of "fixed-position coding", but the introduction and rest of paper don’t describe it and its implementations.

2. Many short form are in abstract (BiLSTM, IQ, AP, BPSK, QPSK, and 16QAM, etc), if possible use full forms with short. 

3. Equation 1 on line 141: verify

4. The difference b/w CGAN and GAN is the condition y?

5. The subsection (2.2. Transformer) is sufficient?

6. Figure 2. Flowchart of OFDM base and communication system: e^( -j2pi ft), is there -ve sign?

7. Table 8. Network structure of Generator Transformer Encoder Layer: beta1 and beta2 are extra large in font.

8. They may separately use title Conclusion, and future work instead of Summary and prospects.

Author Response

(The authors gave the same response as above.)

Reviewer 4 Report

Comments and Suggestions for Authors

This paper introduces a novel Dual Discriminator Conditional Generative Adversarial Network (CGAN) model, combining Long Short-Term Memory (LSTM) and Transformer networks, for the reconstruction of Orthogonal Frequency Division Multiplexing (OFDM) signals. The method has demonstrated satisfactory performance in reconstructing signals in various modulation formats such as BPSK, QPSK, and 16QAM, and has been applied to improve the quality of time series waveforms, constellation diagrams, and spectrograms. It enhances the signal reconstruction process while maintaining a balance between complexity and signal quality. However, there are areas that require improvement, including:

1.         Some of the more technical and complex terms and concepts used in the article, such as "fixed-location coding" and "Vision in Transformer (VIT)", while their role can be generally understood in context, the lack of specific definitions and explanations may confuse readers who are not familiar with these concepts

2.         In Chapter 5, although basic information about the dataset and evaluation metrics for the reconstruction results are provided, the specific details of some experimental settings, such as the basis for choosing noise levels and the process of tuning model parameters, are not explained in detail. This makes it difficult for readers to fully understand the reproducibility and reliability of the experimental results.

3.         In Section 5.2.2, it is suggested to put together the visual comparison diagram of BPSK style, QPSK style and 16QAM style under different SNR. This can more prominently illustrate that as the signal-to-noise ratio decreases, the reconstructed signal becomes increasingly distorted. Nevertheless, the reconstructed spectrogram maintains consistent overall distribution characteristics with the original signal.

4.         Please ensure consistent citation formatting, such as "Ye et al. [13]" on line 69 and "FENG Qi et al. [14]" on line 76.

5.         There should be a space between units and numerical values, for example, lines 229-230 should be "(10 dB, 15 dB, and 20 dB)".

6.         Please check whether the numbering of the tables in the article is consistent. For example, it is not clear what "Table 4.2" in line 308 specifically refers to.

Comments on the Quality of English Language

None

Author Response

(The authors gave the same response as above.)

Round 2

Reviewer 1 Report

Comments and Suggestions for Authors

The revised manuscript addresses my previous comments, and I recommend this manuscript for publication.

Author Response

Thank you for reviewing the manuscript and helping me improve my paper.

Reviewer 2 Report

Comments and Suggestions for Authors

The author has made the necessary modifications as requested

Author Response

(The authors gave the same response as above.)
